# Paclobutrazol Promotes Root Development of Difficult-to-Root Plants by Coordinating Auxin and Abscisic Acid Signaling Pathways in *Phoebe bournei*

**DOI:** 10.3390/ijms24043753

**Published:** 2023-02-13

**Authors:** Jing Li, Peiyue Xu, Baohong Zhang, Yanyan Song, Shizhi Wen, Yujie Bai, Li Ji, Yong Lai, Gongxiu He, Dangquan Zhang

**Affiliations:** 1School of Forestry, Central South University of Forestry and Technology, Changsha 410004, China; 2Key Laboratory of Soil and Water Conservation and Desertification Combating of Hunan Province, Changsha 410004, China; 3Department of Biology, East Carolina University, Greenville, NC 27858, USA; 4School of Forestry, Henan Agricultural University, Zhengzhou 450002, China

**Keywords:** root system, paclobutrazol, IAA, ABA, *Phoebe bournei*, transcriptome

## Abstract

*Phoebe bournei* is a rare and endangered plant endemic to China with higher-value uses in essential oil and structural wood production. Its seedlings are prone to death because of its undeveloped system. Paclobutrazol (PBZ) can improve root growth and development in certain plants, but its concentration effect and molecular mechanism remain unclear. Here, we studied the physiological and molecular mechanisms by which PBZ regulates root growth under different treatments. We found that, with moderate concentration treatment (MT), PBZ significantly increased the total root length (69.90%), root surface area (56.35%), and lateral root number (47.17%). IAA content was the highest at MT and was 3.83, 1.86, and 2.47 times greater than the control, low, and high-concentration treatments. In comparison, ABA content was the lowest and reduced by 63.89%, 30.84%, and 44.79%, respectively. The number of upregulated differentially expressed genes (DEGs) induced at MT was more than that of down-regulated DEGs, which enriched 8022 DEGs in response to PBZ treatments. WGCNA showed that PBZ-responsive genes were significantly correlated with plant hormone content and involved in plant hormone signal transduction and MAPK signal pathway-plant pathways, which controls root growth. The hub genes are observably associated with auxin, abscisic acid syntheses, and signaling pathways, such as PINs, ABCBs, TARs, ARFs, LBDs, and PYLs. We constructed a model which showed PBZ treatments mediated the antagonism interaction of IAA and ABA to regulate the root growth in *P. bournei*. Our result provides new insights and molecular strategies for solving rare plants’ root growth problems.

## 1. Introduction

The majority of rare woody plants have difficulties in root growth, such as low rooting rate, long rooting process, and extraordinarily underdeveloped and vulnerable root systems in the seedling stage. This problem seriously restricts the seedling breeding and natural regeneration of the wild population [1,2]. It is even one of the momentous reasons that lead to the scarcity of rare plants and sporadic distribution in fragile and limited habitats, and finally, face extinction risks [2,3]. Simultaneously, for difficult-to-root rare plants with higher-value uses in essential oil and structural wood production, low transplant survival rate and poor seedling quality are incredibly detrimental to the survival and growth of plants, as well as forest production and cultivation [4]. In addition, roots growing in natural or cultivated soil for a long time are easily affected by weather conditions and physical, chemical, and biological processes related to soil, including the uneven distribution of nutrients, the action of horticultural additives or plant growth regulators, and microbial activities [5]. Accordingly, a vigorous and robust root system (high root biomass, large surface area, and more lateral roots) have positive influences for effectively absorbing and transporting water and nutrients, promoting plant growth and natural regeneration. How to promote root growth of difficult-to-root plants has become the keystone and difficulty in cultivating and protecting these rare plants.

In seedling production, some methods are used to improve root growth, such as selecting and cultivating excellent germplasm resources, strengthening nutrition management, and applying plant growth regulators [6]. However, the breeding and nutrition management of new varieties are time-consuming and require a lot of staffing and other resources. Therefore, adding plant growth regulators becomes a low investment but highly effective seedlings cultivation practice. PBZ (C_5_H_20_ON_3_Cl) is a nitrogen-containing heterocyclic triazole plant growth regulator, which is widely and effectively used in agricultural production and horticultural plant cultivation [7]. Its root growth-regulating property is mediated by altering the balance of essential plant endogenous hormones, including gibberellic acid (GA) and abscisic acid (ABA) [8,9]. However, it is uncertain how paclobutrazol affects auxin to regulate the root system of rare woody plants. More importantly, the species’ biological characteristics and application dose determine the effect of PBZ [9], so it is difficult to explicate the function of PBZ on roots, promoting or inhibiting. For instance, PBZ at low concentrations (10 mg/L) could improve the drought resistance of plants by increasing the relative growth rate of roots and the contents of ABA, GA_3_, IAA, and zeatin (ZT) in *Amorpha fruticosa* [10]. However, 60–80 mg/L of PBZ inhibited the root growth of perennial ryegrass (*Lolium perenne*) [11]. The molecular mechanism is also rarely discussed due to its high complexity and diversity. Therefore, it is paramount to clarify the response mechanism to the concentration of PBZ in roots to understand that PBZ regulates root growth by mediating plant endogenous hormones and guiding the cultivation of rare plants with underdeveloped roots.

Plant endogenous hormones affect root structure by monitoring and reacting to environmental alterations and availability of nutrients, such as primary root elongation and the increase of lateral roots [12,13]. Increasing evidence indicates that plant endogenous hormones build a complex regulatory network interacting with each other’s synthesis or response pathways, which mediate root growth and development [14]. Among these endogenous hormones, IAA and abscisic acid are crucial in the entire physiological process of root growth and development [13,15].

Auxin is requisite for regulating the development and growth of roots. Many plants modulate auxin content to adjust root structure in response to environmental changes and improve plant adaptability [16]. Multiple studies have demonstrated that the modulation of auxin on root structure is connected with the asymmetric spatial and temporal pattern of auxin distribution in root tissues [17]. This process is supported by the polar transport carrier proteins on the cell membrane, primarily the auxin export carrier proteins such as PINs and ABCBs. These genes affect root formation by controlling polar auxin transport (PAT) between cells and generating a suitable concentration gradient [17]. Auxin content is affected not merely by the auxin transport but also by the processes of auxin synthesis and decomposition, ABA, and GA_3_ content [18]. As an illustration, in *A. thaliana*, auxin synthesis genes (such as *ASA1* and *TAR2*) and GH3s, AU X/IAAs, and SAURs auxin response genes can influence root traits, including the length of primary and lateral roots, and the number of lateral roots [19,20].

As a “stress hormone”, ABA could also help plants adapt to environmental changes by regulating root growth and development [15,21]. Many studies have shown that ABA signal transduction components are directly or indirectly involved in plant root development and morphogenesis, notably PYR/PYLs (pyrabactin resistance/pyr1-like components of ABA receptor), PP2Cs (protein phosphatase 2C), SnRK2 (SNF1-related protein kinase 2), and ABFs (ABRE binding factor) [21]. Additionally, ABA influences auxin homeostasis in roots by regulating the expression of genes associated with IAA synthesis and signal transduction (YUCs and PINs) [22], as well as regulating root development in cooperation with auxin signaling [23]. Moreover, the crosstalk exists between ABA and IAA signaling pathways, and ABA usually affects lateral root growth by antagonizing auxin. For instance, in *Arachis hypogaea*, ABA inhibited auxin transport to limit IAA in generating lateral root primordia [24]. Excessive auxin content promoted the differentiation of distal stem cells, whereas the required ABA contents were the contrary [25]. As a corollary, IAA and ABA have a dynamic interaction that could be antagonistic or synergistic in plant roots. This dynamic relationship is necessary for plants to modulate root structure in order to adapt to changing conditions and their growth and development. There are, nevertheless, few investigations on the dynamic synergistic or antagonistic interaction between them and the related regulatory mechanism, and there are no reports in uncommon, difficult-to-root rare plants.

*Phoebe bournei* is a difficult-to-root plant of Phoebe, Lauraceae. It is a typical example of China’s unique, rare, and endangered woody plants [26]. Currently, natural populations are rarely scattered in Fujian, Zhejiang, Hunan, and other provinces in southern China due to recruitment limitations (seed and seedling stage) and human deforestation [26]. *P. bournei* is chosen as a precious wood tree species in southern China owing to its straight trunk, resistance to corrosion, and resilience to dampness. In addition to the unique scent of *P. bournei*, it endows important industrial production values such as wood decay resistance and high-quality essential oil extraction [27,28]. As a result, *P. bournei* has been designated as a species of forest tree that necessitates a national strategic reserve, and it is imperative to protect wild resources and cultivate effective seedlings. Although the challenge of rooting *P. bournei* cuttings has been partially resolved, the survival rate is incredibly low due to poor seed germination, weak roots in seedlings that have sprouted, and quick demise [29]. Presently, *P. bournei* plantation cultivation and the regeneration of the natural populations still depend on seed propagation [29]. However, the unresolved difficulties with seed germination and root growth severely restrict the seedlings’ production and continue to endanger the natural forest regeneration of *P. bournei*. Therefore, in this study, we analyzed the root architecture and dynamic changes of IAA and ABA, the transcriptional expression patterns of paclobutrazol-responsive genes in roots under different PBZ concentrations, in aiming to unveil the physiological and molecular mechanisms by which PBZ regulates root growth.

## 2. Results

### 2.1. Moderate-Concentration Paclobutrazol Promoted Root Growth and Development in P. bournei

PBZ affected root growth and development in a dose-dependent manner in *P. bournei* seedlings (Figure 1A). Compared with the controls (NT), all tested root traits (TRL, SA, and NLR) were first increased and then decreased subsequently with increasing PBZ concentrations (Figure 1B–E). In particular, the TRL, SA, and NLR were significantly increased under the MT (*p* < 0.01) (Figure 1B–E). They were 69.90%, 56.35%, and 47.17% greater than the control. In addition, the TRL in the MT was significantly higher than the LT and HT by 47.29% and 61.29%, respectively. However, there was no significant alteration in primary root length among different treatments (Figure 1D).

### 2.2. Paclobutrazol Induced Genome-Wide Aberrant Expression of Genes in P. bournei

To further investigate the potential regulatory mechanism caused by PBZ treatments in difficult-to-root plants, we employed RNA-seq and qRT-PCR to analyze genome-wide the transcriptome profiles in the roots of *P. bournei* treated with different concentrations of PBZ (Figure 1, Appendix A). The qRT-PCR results were consistent with the DEG profiling (*p* < 0.05, Appendix A). The linear regression analysis of RNA-seq results (FPKM) and qPCR results (2^−ΔΔCt^) showed a significantly positive correlation (R^2^ = 0.808, NT vs. LT; R^2^ = 0.852, NT vs. MT; R^2^ = 0.826, NT vs. HT) (Appendix A). The above results indicated that the RNA-seq data were credible and accurate. A total of 8022 DEGs were identified after PBZ treatments (Figure 2A–C). The results showed that 45.89%, 25.75%, and 28.36% of the DEGs were detected in roots treated with LT, MT, and HT, respectively (Figure 2A–C). Different PBZ treatments caused a different impact on gene expression profiles. In particular, we observed the number of up-regulated and down-regulated DEGs was similar (49.47% and 50.53%, respectively) in the MT (Figure 2B). The up-regulated DEGs (38.49% and 42.95%) were significantly less than the down-regulated DEGs (61.51% and 57.05%) in the LT and HT, respectively (Figure 2A,C). It indicated that low and high concentrations of PBZ have a stronger inhibitory effect on the genes in the roots. Moreover, there were significant differences in the number of DEGs among paclobutrazol treatments (Appendix A). The number of up-regulated DEGs (64.20%) was greater than the down-regulated DEGs in the MT (35.80%) when compared with the LT and HT (Appendix A). The aforementioned findings suggested that the moderate concentration of PBZ had a promoting effect on the expression of DEGs in roots. In addition, Venn diagrams analysis revealed that the specific paclobutrazol responsive gene (675, 8.41%) was more than that of the common (4752, 59.24%) (Figure 2D). Among them, only in the moderate PBZ concentration were the specific up-regulated DEGs (338) more than the down-regulated DEGs (239) (Figure 2D). So based on the above process, PBZ strongly affected gene expression in the roots in a dosage-dependent manner.

Differentially expressed genes induced by PBZ were mainly involved in carbon metabolism, signal transduction, transport and catabolism, and environmental adaptation pathways (Figure 2E). Compared with the NT, the genes responding to the PBZ treatments were mostly enriched in metabolic pathways, MAPK signaling pathway-plant, and plant hormone signal transduction pathways (Figure 3B). There are three exceptions, respectively, “glutathione metabolism”, “glycine, serine, and threonine metabolites”, and “zeatin synthesis”, which were more enriched under MT and HT than under other treatments in the roots (Figure 3B). In addition, there were differences between the treatment groups in the “plant hormone signal transduction”, “α-linolenic acid metabolism”, “linoleic acid metabolism”, and “ABC transporters” (Figure 3B).

Gene Ontology (GO) analysis showed that the pathways regulated by PBZ treatments were abundant and enriched in the “metabolic process”, “signaling”, “transporter activity”, “cell”, and “membrane” (Appendix A). The number of up-regulated DEGs under medium concentration was more than down-regulated among the top 20 GO terms (Appendix A). A further study shows that the GO terms, such as “auxin-activated signaling pathway”, “hormone biosynthetic process”, “L-phenylalanine metabolic process”, “active transmembrane transporter activity”, “peptide transmembrane transporter activity”, and “ion transmembrane transport”, were more significantly enriched under the medium concentrations than other treatments (Figure 3A). Additionally, some GO terms related to transmembrane transport also existed only in the moderate and high concentration, including transmembrane transport, peptide transporter activity, and amide transmembrane transporter activity (Appendix A). These results revealed the difference in concentration-specific outputs of the transcriptomes in the roots under different PBZ treatments; the medium concentration responsive genes are significantly correlated with plant hormone signal transduction and transmembrane transport pathways.

To determine the paclobutrazol-specific responsive genes associated with root growth, we identified 11 modules containing 18,711 genes using WGCNA (Figure 4A,B, Appendix A). Among them, the turquoise module included more genes (2629) (Figure 3A and Appendix A). Furthermore, these genes were significantly positively correlated with the total root length (r = 0.46, *p* ≤ 0.000), number of lateral roots (r = 0.72, *p* ≤ 0.000), the total lateral root length (r = 0.71, *p* ≤ 0.000), auxin concentration (r = 0.78, *p* ≤ 0.000) (Figure 4B), and negatively correlated with ABA concentration (Figure 4C). Genes in the turquoise module were over-represented in these pathways: ribosome, plant hormone signal transduction, MAPK signaling pathway-plant, and phenylalanine anabolism. Among them, more genes were related to root traits in the plant signal transduction pathway (Figure 4D). We used the CytoHubba application in Cytoscape to visualize gene regulatory networks and further mine the key genes. We predicted a few hub genes (weight > 0.01, the top 25 in degree value), including *ABCB28* (MSTRG.15088), amino acid transferase (*TAR3*, MSTRG.22058), *MYB23* (MSTRG.23221), *PIN4* (Maker00036823), and defense-related genes (MSTRG.11953, MSTRG.24054, and MSTRG.21263) (Figure 4E, Appendix A). Further studies conclusively showed the hub genes cloud relates to the following genes: containing auxin (20), abscisic acid (10), gibberellin (4) synthesis and signal transduction-related genes, root growth-related genes (10), MYBs, WRKYs, and other transcription factors (20) (Figure 4E). Based on the preceding, we consider that plant hormones, in particular auxin and abscisic acid, could play an essential role in regulating root growth by PBZ.

### 2.3. Paclobutrazol Regulates Root Growth and Development by Mediating IAA and ABA Biosynthesis and Signal Transduction in P. bourne

RNA-seq analysis suggests that many DEG genes were associated with auxin and ABA biosynthesis and signaling pathways (Figure 5). Differential expression genes involved in the IAA signaling pathway were identified in all four PBZ treatments (Figure 5D). Auxin synthesis genes, *ASA1* (anthranilate synthase alpha subunit 1) (Maker00001235, Maker00011703), *ASB1* (anthranilate synthase β1) (Maker00003680), *IGPS* (indole-3-glycerol phosphate synthase, auxin synthesis pathway-glycerol phosphate synthase) (Maker00021486), *TSA1* (tryptophan synthase subunit alpha subunit 1) (Maker00046755), *TSB1* (tryptophan synthase subunit β subunit 1) (Maker00014316, Maker00053345) were upregulated by PBZ (Figure 5D). Differentially, the expression of most of these auxin synthesis genes was the highest in MT. In the IPA (Indole-3-pyruvate) auxin synthesis pathway: TARs (Tryptophan aminotransferase) (Maker00001235, Maker00011703, Maker00003680) were upregulated in MT and were upregulated in other treatments (Figure 5C). In the YUCCA pathway (flavin monooxygenases), *YUC2* (Maker00003172) and *YUC10* (Maker00044572) were also induced in the MT but significantly inhibited in NT and HT treatments (Figure 5C). In the indole acetamide auxin synthesis pathway (IAM), *AAO1* (Maker00025222) and *AMI* (Maker00004829) were induced by the HT. In addition, *PIN4* (Maker00036823), *PILS7* (Maker00025932), and *LAX3* (Maker00013705) were significantly upregulated in MT (Figure 5D). The transcript abundance of genes in IAA synthesis and signaling changed dynamically as the concentration of PBZ treatments corresponded to the differences in IAA content in the roots (Figure 5A). The medium concentration of PBZ significantly increased the auxin content in the roots (*p* < 0.01), which was 3.84 times, 1.87 times, and 2.47 times higher than that of the control, low, and high concentrations, respectively (Figure 5A). In addition, IAA content was 68.88%, 39.34% higher at low and high PBZ concentrations than in the control group.

The F-box protein *TIR1* (Maker00038877) and *AFB2* (Maker00037139) were upregulated in the MT but downregulated in HT. Seven early auxin response genes AUX/IAAs (Maker00052669, MSTRG.12062, MSTRG.26774, MSTRG.5823, Maker00055260, Maker00056424, and Maker00033700) were downregulated with the increase of PBZ concentrations (Figure 5D). Nine ARFs regulating auxin response genes were differentially expressed: five (Maker00025936, Maker00036351, Maker00019632, Maker00017991, and MSTRG.3396) were upregulated in LT, and four (Maker00020047, MSTRG.26950, Maker00034333, and Maker00026481) were strongly upregulated in MT (Figure 5D). Additionally, in comparison with other treatments, the auxin-responsive genes, including GH3s (Maker00052834, Maker00042207, Maker00016041, Maker00016231, and Maker00006967), SAURs (small auxin-up RNA) (Maker00026652, Maker00033302, and Maker00033652), AUX/IAAs (Maker0002102, Maker00024832, and Maker00033264) were significantly up-regulated in MT (Figure 5D). Five LBDs (lateral organ boundaries domain) (Maker00033119, Maker00038035, Maker00040821, Maker00029955, and Maker00011369) and the target genes of the plant hormone auxin regulators (ARFs) were significantly upregulated by MT but strongly inhibited by HT (Figure 5D). These results showed that PBZ significantly affected the auxin biosynthesis and signaling pathway in the roots of *P. bournei*. Specifically, medium concentrations of PBZ could significantly enhance most of the gene expression related to the auxin signaling pathway (Figure 6).

PBZ treatments significantly inhibited the expression of ABA synthesis-related genes, including *PDS* (Maker00013437), *VDE1* (violaxanthin de-epoxidase) (Maker00022568), SDRs (MSTRG.17972, MSTRG.948, Maker00021357, and Maker00028444), and the expression level decreased with the increase of PBZ concentrations. ZEPs (zeaxanthin epoxidase) (Maker00038588, Maker00038590, and Maker00051987) were upregulated in MT and HT (Figure 5E). Meanwhile, NCEDs (Maker00024115, Maker00026779, and Maker00031262) were upregulated in the PBZ treatments. Notably, the expression of NCEDs was maintained at a relatively low level in the roots (Appendix A). CYP707As (Maker00043915 and Maker48092) mediating the degradation process of ABA were also upregulated in the LT and MT (Figure 5E). This result coincided with changes in the content of ABA under different treatments in the roots (Figure 5B,E). The ABA content in roots treated with different concentrations of PBZ was markedly less than that of the NT (*p* < 0.01). It decreased by 47.78%, 68.84%, and 34.58% at low, moderate, and high concentrations of PBZ, respectively. Among them, the content of ABA that was available at moderate concentration decreased by 30.84% and 44.79%, respectively, compared to low and high PBZ concentrations (Figure 5B and Figure 6). In addition, the moderate concentration of PBZ inhibited most genes in the ABA signaling pathway, including SnRKs (MSTRG.6787, Maker00001687, and Maker00039349), *ABI5* (Maker00012439, Maker00026069, Maker00048786, Maker00014859, Maker00037370, and Maker00014147) (Figure 5E and Figure 6). They are the positive regulator in the ABA signal transduction. PP2Cs (Maker00026758, Maker00045445, Maker00027116, and Maker00039030), inhibiting ABA signaling and promoting GA signaling, were increasingly upregulated over the PBZ concentrations (Figure 5E). In particular, MT treatment significantly promoted the expression of PYR/PYLs in roots (Maker00023439, Maker00013172, Maker00027195, and Maker00028633) (Figure 5E and Figure 6).

According to the results presented above, the moderate concentration considerably boosted the expression of genes involved in IAA synthesis, which contributed to a significant increase in the contents of IAA released, promoting the auxin signaling in *P. bourne* roots. However, the moderate concentration of PBZ also severely reduced the contents of ABA available, as well as the biosynthesis and signal transduction of ABA (Figure 6).

## 3. Discussion

PBZ is an effective plant growth regulator commonly utilized in agricultural and horticultural cultivation. However, few studies have deciphered its effects on the roots of rare plants in the horticultural and ornamental fields, and its regulation mechanism is unclear. Aimed at the rare plants with difficult-to-generate lateral roots, we used *P. bournei* to investigate the impact of PBZ and its regulatory mechanism on root growth and development at both physiological and molecular levels and to elucidate the interaction mechanism of the plant endogenous hormones in response to PBZ treatment. This study could open up avenues and provide the scientific foundation for deciphering how plant growth regulators promote the root growth of horticultural and ornamental rare plants.

### 3.1. Paclobutrazol Enhances Root Growth and Development by Promoting IAA Biosynthesis and Signaling Pathway in Difficult-to-Root Rare Plants

Previous studies have mainly interpreted PBZ as an inhibitor of gibberellin and that it regulates and affects the growth and development of model plants such as rice and *A. thaliana* [7]. However, it is still unknown how PBZ influences IAA contents and regulates root development in rare woody plants. This study explains for the first time that medium concentrations of PBZ activated auxin biosynthesis and its signaling pathway, increased auxin content, and subsequently positively impacted the growth of roots in rare woody plants. Our results would provide new insights into the possible underlying mechanism of PBZ and a practical method for solving the problem of undeveloped root growth in rare woody plants.

Increased PBZ concentration gradually activated the IAA biosynthesis and signaling pathway at the transcriptional level. Most IAA-related genes were highly expressed at medium concentrations and drastically receded at high concentrations of PBZ (Figure 5C). Meanwhile, IAA content also increased significantly at medium concentrations. All these indicated that IAA synthesis and its signaling pathway participate in PBZ regulating root growth and development, and medium concentration significantly facilitated this process. As the predominant endogenous form of auxin, the plant hormone IAA could control cell division and elongation to regulate the primary and lateral root growth by PAT, forming asymmetric distribution and concentration gradients in roots. The auxin synthesis pathway depends significantly on the *TSB1* and *ASA1* genes. Their upregulated expression could raise the content of IAA and promote the growth of lateral roots in *A. thaliana* [30,31]. *TSB1* is also involved in maintaining a homeostatic balance of IAA and ABA to regulate root growth and development [31]. These genes were extensively upregulated by PBZ treatment in this study. Therefore, we speculated that PBZ treatment could promote IAA synthesis-related genes, increasing the IAA contents and triggering root growth and the formation of lateral roots.

According to the polar auxin transport chemiosmotic hypothesis, auxin molecules are deprotonated in the more alkaline cytosol, which would lead to anions that only pass the plasma membrane with the assistance of auxin efflux carriers to finish cell-to-cell polar transport in roots [17]. Studies have demonstrated that the increase of IAA content induced the expression of PINs [32], facilitated the flow of auxin to the root tip, root cap, root epidermis, and other tissues, and then positively regulated root growth. The high expression of PIN genes also upregulated the expression of essential proteins associated with auxin biosynthesis and signal transduction, such as *TSB1*, TARs, ARFs, and LBDs, which, in turn, strengthened root formation and growth [17,33]. In this study, we also observed the increased expression of these genes at medium concentrations of PBZ. Thus, we conclude that the application of PBZ at medium concentrations could improve root growth by promoting auxin biosynthesis and transport.

High auxin concentrations could also activate a classical Aux/IAA-TIR1-ARFs signal transduction pathway [16]. High-concentration auxin binds to TIR1/AFB nuclear receptors and facilitates Aux/IAA ubiquitination and degradation by SCFTIR1/AFB-proteosome module [20]. Released ARFs increased (Figure 5C) and formed dimers and even higher-order complexes to promote the expression of downstream target genes, such as GH3s, AUX/IAAs, SAURs, and LBDs. Among them, LBDs are the lateral organ boundaries domain transcription factor, which promotes lateral root growth [20,34]. In contrast, low concentrations of auxin did not initiate this process [20]. Previous research has revealed that these genes were promoted by ARFs, such as *ARF7* and *ARF19*, in *A. thaliana* and then participated in the asymmetric division of primary root and lateral root cells to control the formation of roots [34]. In our research, the medium concentration of paclobutrazol promoted the expression of these genes (Figure 5D). Interestingly, we found a significant association between paclobutrazol-induced hub genes, ARFs, and LBDs (Figure 4E). All of this leads us to conclude that the whole auxin synthesis, transport, and signal transduction system was involved in the beneficial effect of PBZ on the rooting of difficult-to-root rare woody plants.

### 3.2. Paclobutrazol Inhibited ABA Biosynthesis and Signaling Pathway in the Roots of Difficult-to-Root Rare Plants

As increasing PBZ concentrations, ABA contents were first substantially decreased and then increased in *P. bournei* roots. This dynamic fluctuation was closely related to ABA biosynthesis and metabolism [35]. PBZ treatments first induced the expression of NCEDs and ZEPs and subsequently significantly decreased the expression of PDSs and SDRs. SDR is a pivotal enzyme in ABA biosynthesis during the transformation of Xanthoxin (XAN) to ABA [36]. Therefore, its down-regulation would directly inhibit ABA synthesis. In addition, *CYP707A* is an integral gene in the process of ABA catabolism. It could encode the ABA 8’-hydroxylase in the ABA oxidative inactivation pathway. Scientists have already shown that this gene can regulate the ABA negative feedback process, enabling plants to attain the best growth [37]. In this study, we detected an increase in the expression of CYP707As at low and medium concentrations. PBZ, which is most likely acting as a stressor, induced the expression of genes NCEDs and ZEPs, activated the ABA negative feedback regulatory mechanism, and whereat decreased the concentration of ABA. Moreover, the reduction in the concentration of ABA may also associate with the increase of GA_3_ in *P. bournei* roots. ABA and gibberellins have a common terpenoid biosynthesis pathway [7]. After treatments with PBZ, the GA_3_ content was increased in the roots (Figure 5C), indicating that fewer terpenoid precursors were accumulated and shunted to the ABA synthesis process [7]. As a corollary, the ABA contents were decreased in the roots of *P. bournei*.

Abscisic acid has different effects on root regulation in a dosage-dependent manner. At low concentrations, abscisic acid promotes root growth; but at high concentrations, it inhibits root development [38]. This regulatory mechanism is closely related to abscisic acid signal transduction components. The decrease in abscisic acid content promotes the expression of the PP2C genes and dephosphorylation of SnRK2 genes and blocks the ABA signal transduction process [38]. In this study, paclobutrazol treatment significantly inhibited the expression of genes involved in abscisic acid signal transduction in the roots of *P. bournei*, such as the genes *SnRK2* and *ABI5*. Related studies have shown that the abscisic acid-responsive gene *ABI5* has a negative regulatory effect on root growth [39], and SnRK2 genes are also significantly associated with root growth [40]. In addition, the up-regulated expression of PYLs in *A. thaliana* promoted the growth of lateral roots [15], which is consistent with the results of this study.

### 3.3. Interaction of ABA and IAA in Response to Paclobutrazol Treatments

The complicated interactions between auxin and ABA in regulating root development are dependent on their content and environmental changes [15], which are regulated majorly by the differential expression of core genes involved in plant hormone response [12]. This has been well-studied in herbaceous plants such as *A. thaliana*, soybean, and rice [15,31]. However, there is no research on how paclobutrazol mediates the antagonistic relationship between auxin and abscisic acid in woody plants. In this study, we found that a medium concentration of paclobutrazol inhibited the expression of ABA-related genes, such as PDS, SDRs, and *ABI5*, and ABA content was decreased while PINs were up-regulated. In *A.thaliana*, ABA reduced auxin content in roots, leading to root growth retardation, which is related to the expression of PIN genes that were inhibited in roots by the high concentration of ABA [41]. Furthermore, *ABI4/ABI5* reduced auxin content in *A. thaliana* roots by restraining the accumulation of PINs, thereby inhibiting root meristem growth [39]. Due to the above reasons, we speculate that paclobutrazol inhibits ABA synthesis and transduction-related genes, relieving its inhibition of auxin and promoting auxin synthesis and signaling in roots. All of these promoted root growth and development ultimately.

PYLs could interact with MYBs to enhance the expression of ARFs and downstream LBDs to participate in the positive regulation of auxin biosynthesis on *A. thaliana* lateral roots [42]. Our results showed that only PYLs were up-regulated in ABA synthesis and signaling pathways after a medium concentration of PBZ treatment. MYBs, ARFs, and LBDs were also significantly up-regulated. At the same time, WGCNA analysis confirmed that the PBZ responsive hub genes, such as *TAR3* [19], PIN4, and MYBs, were significantly correlated with IAA and ABA-related genes regulating root growth (like LBDs and ARFs) (Figure 4E). Therefore, we thought that PBZ might be involved in the regulation of root growth by acting on the hub genes of ABA and IAA interaction. Additionally, PBZ induced the expression of a large number of MAPK-related genes, such as MAPKKKs (mitogen-activated protein kinase kinase kinase). According to previous studies, ABA promoted root growth by interacting with auxin, MAPK signaling pathways, and cell-cycle machinery [43]. We thus speculated that the interaction between IAA and ABA could also be related to MAPK signal transduction during PBZ treatments.

### 3.4. Roles of GA_3_ in Root Growth and Response to Paclobutrazol Treatment in the Difficult-to-Root Rare Plants

Unlike previously published research results, PBZ treatments increased GA_3_ content in our result (Figure 5C). Transcriptome data showed that except for GA20ox, the expression patterns of other vital genes of gibberellin synthesis were inconsistent with the changes in GA_3_ content, and medium and high concentrations of PBZ inhibited the expression of genes related to GA_3_ synthesis and signal transduction (Figure 5F). The biosynthesis and metabolism of GA_3_ is a multi-step enzymatic reaction involving many important enzymes. Environment signals affect GA_3_ synthesis and metabolism by precisely regulating the expression of critical enzymes, such as GA20-oxidase, GA2-oxidase, and GA3-oxidase, in which complex feedforward and feedback mechanisms provide the possibility for this fine regulation [44,45]. The feedback mechanism would control the expression of GA3ox and GA20ox depending on whether the GA_3_ concentration is too high or low [46]. For instance, GA_3_ could induce the expression of GA20ox2 and GA3ox genes, increase endogenous gibberellin content, and then promote stem growth [47]. However, under certain conditions, GA2-oxidase could catalyze bioactive gibberellins into inactive gibberellins [46]. In addition, other plant endogenous hormones can also affect gibberellin synthesis and signaling through crosstalk between hormones. IAA could increase GA_3_ content and promote GA metabolism by upregulating the expression of GA synthesis-related genes (such as GA3ox and GA20ox) and downregulating GA catabolism-related genes, such as GA2ox, to regulate plant growth [48]. However, ABA inhibited the expression of GA20ox2 and GA3ox2 genes and reduced the content of active GA in plants. In addition, in this study, after PBZ treatments, the content of IAA was increased, but ABA was decreased, which might induce GA20ox expression and affect the GA_3_ content. Additionally, PBZ concentrations and species differences might also be the reasons our results differed from those of other researchers. Based on the above results, we thought that PBZ inhibited essential genes in gibberellin synthesis but simultaneously triggered complex negative feedback regulation of gibberellin. This process might be related to ABA and IAA content and the genes in the GA synthesis pathway, such as GA2ox and DELLA protein.

## 4. Materials and Methods

### 4.1. Plant Growth and Paclobutrazol Treatment

Seeds were collected from a 15-year-old *P. bournei* tree grown in Jindong State Forest Station (112°04′30″ E, 26°18′30″ N), Hunan province, China. Through sand storage, the seeds were sterilized in 75% (*v*/*v*) alcohol and then washed with distilled water five times. After germination on filter paper in the dark at 25 ± 2 °C for 96 h, we transferred 240 seedlings to PVC columns (height 10 cm and diameter 10 cm, one seedling per hole) filled with sterilized river sand. These seedlings were irrigated with 1/2 Hoagland nutrient solution (Hoagland’s, Coolaber, China) and grown in a plant room with 28 ± 2 °C (day)/23 ± 2 °C (night), 50–70% relative humidity, and 16 h/d light/dark. We replaced the nutrient solution every two weeks to maintain the same concentration. After 12 weeks, the seedlings were treated with different concentrations of PBZ by soil irrigation. The concentrations were 0, 0.6, 6, and 60 mg/L, respectively, referred to as low (LT), medium (MT), and high concentrations (HT) according to their relative dosage. These concentrations were selected based on our previous field experiments, which were safe and effective concentrations for the seedlings [7].

### 4.2. Analysis of Root Morphological Traits

Following the paclobutrazol treatments that lasted for 14 days, 50 seedlings were chosen randomly from each group. Of those seedlings, 15 were scanned for roots (Expression 10000XL 1.0, Epson Inc, Suwa, Nagano Prefecture, Japan), and the results were evaluated using the WinRhizo Pro (S) v.2004b software (Regent Instruments Inc, Sainte Foy, QC, Canada). The measured root traits included the total root surface area (SA), the total root length (TRL), the primary length (PL), and the lateral root number (NLR). The roots were collected, immediately frozen in liquid nitrogen, and then stored at −80 °C for subsequent experimental analysis.

### 4.3. Endogenous Hormones Quantification

We used high-performance liquid chromatography-electrospray ionization mass spectrometry (ESI-HPLC-MS/MS) (triple quadrupole-ionic hydrazine mode, mass spectrometer Qtrap6500, Aglient 1290, Agilent Technologies Inc., Santa Clara, CA, USA) to detect the contents of IAA, ABA, GA_3_, trans-Zeatin (TZ) and Brassinolide (MeJA) (The standards purchased from Olchemim company, Olomouc, Czech Republic), and the purity was greater than 99%.

For each sample, one gram of fresh roots was sampled and added to isopropanol-water-hydrochloric acid mixed extract after grinding. We added 8 μL of 1 μg/mL internal standard solution; the homogenates were centrifuged at low temperature (4 °C for 30 min) and then added to dichloromethane and oscillated at low temperature (4 °C) for 30 min again. Next, it was centrifuged at a low temperature (4 °C) for 5 min at 13,000 r/min and taken out of the lower organic phase. Ultimately, the supernatant was centrifuged at 4 °C for 10 min (13,000 r), passed through a 0.22 μm filter, and measured by LC-MS/MS with a reversed-phase Poroshell 120 SB-C18 column (2.1 × 150, 2.7 μm, Agilent, USA). The mobile phase was set as A: B = (methanol/0.1% formic acid): (water/0.1% formic acid). The sample injection was 2 μL, the column temperature was 30 °C, and the volume flow rate was 1.0 mL/min. IAA, ABA, GA_3_, TZ, and MeJA were detected using the electrospray anion source (ESI-source). The hormone peak area was measured and compared with the standard curve for calculating hormone concentration. Three biological replicates were run for each treatment.

### 4.4. RNA-Sequencing and DEGs Function Analysis

About 200 mg of root sample was subjected to RNA extraction from each sample that was used in RNA-seq analysis. Three biological replicates were performed for each treatment. Total RNAs were isolated from the roots of the treated seedlings and the controls using the RNAprep Pure Plant Kit (Tiangen, Beijing, China). RNA quality was evaluated using an Agilent 2100 Bioanalyzer (Agilent Technologies, Santa Clara, CA, USA). Twelve cDNA libraries were established using AMPure XP beads (concentration of each sample > 20 ng/μL). The libraries were qualitied by an Agilent Bioanalyzer 2100 and sequenced on Illumina HiSeq (Nova seq 6000, Illumina, Inc, San Diego, CA, USA) platform.

The raw data were first filtered using fastp to remove the joint sequence, and low-quality reads for obtaining the high-quality clean data. The clean reads were compared to the reference genome (https://ngdc.cncb.ac.cn/gsa (accessed on 18 September 2021)) through HISAT2.2.4 [49]. The mapped reads were assembled using the StringTie v1.3.1. The FPKM value was used to show the expression level of different genes. The correlation between different samples was shown by the Pearson correlation coefficient square (R^2^). DESeq2 was used to analyze the differential gene expression between the control and the treatment groups. The *p*-value calculated by the negative binomial distribution model was corrected by multiple hypothesis tests to get access to the FDR value (false discovery rate). Finally, genes with log_2_|FC (Fold change)| ≥ 1 and q < 0.05 were selected as significantly differentially expressed genes. We used the “Complex Heatmap” R package (version 4.1. 0) to visualize the changes in gene expression.

Kyoto Encyclopedia of Genes and Genomes (KEGG, http://www.kegg.jp) (accessed on 13 October 2021) and Gene Ontology (GO, http://www.kegg.jp) (accessed on 28 October 2021) were used to analyze the enrichment of transcripts and DEGs in roots treated with PBZ. KEGG is able to speculate on the functions of protein interaction networks in various cellular activities. GO enrichment analysis could identify the primary biological functions of DEGs. These biological functions include molecular functions, cellular components, and biological processes. The hypergeometric distribution was utilized in order to discover the pathway and GO term, which significantly enriched DEGs in comparison to the genome background. The calculated *p*-value is corrected to obtain FDR, with FDR ≤ 0.05 as the threshold value. The DESeq (2012) R package was used for the hypergeometric distribution.

### 4.5. Weighted Gene Co-Expression Network Analysis

The co-expression network was constructed using the R package WGCNA (v1.63). All DEGs are used to generate an adjacency matrix of pairing relationships. Selecting the soft threshold of co-expression network clustering based on R^2^ > 0.9, the soft threshold of samples was set to 10. The FPKM values were transformed into a topological overlap matrix (TOM) and performed hierarchical cluster analysis. Different genes were classified into co-expression modules by the dynamic branch-cutting approach. Each module’s eigengene value was calculated using the eigengene function, which was used to test the association with root traits. The gene network was visualized using Cytoscape (v3.9.1, National Institute of General Medical Sciences, Bethesda, MD, USA).

### 4.6. qRT-PCR Validation of DEGs

RNA-seq data were validated by qRT-PCR on an ABI StepOnePlus Real-Time PCR equipment. A total of 18 genes were randomly chosen for this validation (Appendix A). Total RNAs were extracted from three biological replicates for each treatment of roots (*n* = 3) in a previous step, which was utilized for reverse transcription. qRT-PCR was performed using SYBR Green dye (SYBR ^®^ Green Real-time PCR Master Mix-Plus, Code No. QPK-212). A total of 20 μL volume was used for each reaction, which included 10 μL 2 × SYBGEEN PCR mix, 1.6 μL forward and reverse primer (10 μ mol/μL), 2 μL cDNA, 0.4 μL ROX, and 6 μL ddH2O. The reaction system was pre-denatured at 95 °C for 2 min, denatured at 94 °C for 10 s, annealed at 60 °C for 10 s, and extended at 72 °C for the 40 s. DREB1a (acc.no.: KX682036) and DREB2b (acc.no.: KX682037) were used as reference genes [50], and the relative expression of genes was calculated by the 2^−ΔΔCt^ method. Calculations were made on the fold change in the relative expression of these genes, as well as the DGE spectra, and a linear regression analysis was carried out.

### 4.7. Statistical Analysis

The R programming language (v 4.2.2) was employed to do statistical analysis and graphing. Duncan’s one-way analysis of variance (ANOVA) was used in order to examine the differences in root traits and endogenous hormone content that resulted from the treatment. The difference was statistically significant when *p* ≤ 0.05.

## 5. Conclusions

In conclusion, we comprehensively and deeply analyzed the physiological and molecular mechanisms by which different PBZ concentrations regulated the root growth of difficult-to-root rare plant seedlings. We revealed that a moderate concentration of PBZ promoted auxin biosynthesis and signal transduction processes, inhibited abscisic acid biosynthesis and signal transduction, and increased the root surface area, total root length, and lateral root number. At the same time, it was presumed that the essential genes, including PINs, ABCBs, TARs, ARFs, LBDs, CYP707As, PYLs, and ABI5, were involved in the control of root growth and development by PBZ. For the first time, our study proposed that the moderate concentration of PBZ promoted root growth and development by mediating the antagonism interaction of IAA and ABA. It provides new insights into how plant hormones interact to respond to the changes in PBZ concentrations to modulate root architecture and affect large-scare root gene expressions, which are considerable for us to understand how plants adapt to the environment and the cultivation and protection of rare plants.

## Figures and Tables

**Figure 1 ijms-24-03753-f001:**
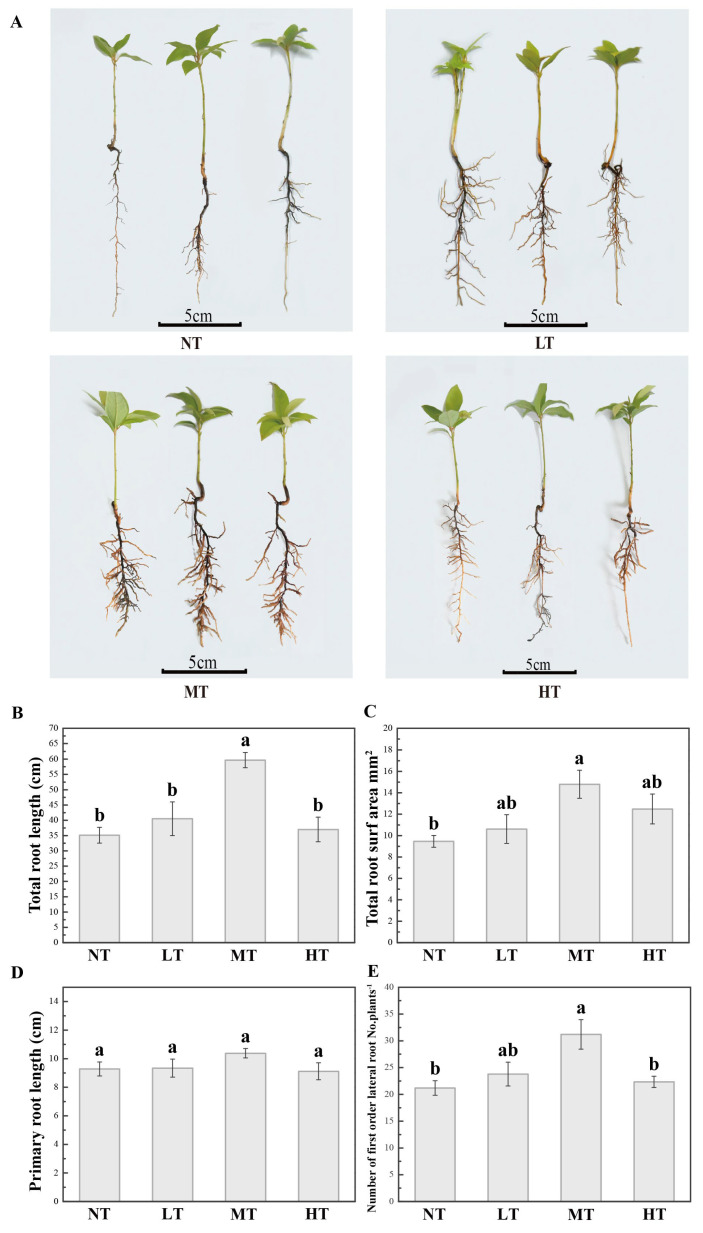
The medium concentration of paclobutrazol (PBZ) significantly promoted root morphological traits in *P. bournei*. (**A**) Three-month-old seedlings grown under control (NT), 0.6 mg/L (LT), 6 mg/L (MT), and 60 mg/L (HT) treatments of PBZ. (**B**–**E**) Root traits, including total root length (TRL), total root surf area (SA), primary root length (PRL), and lateral root number (NLR), respectively, under different treatments of paclobutrazol. Different lowercase letters indicate significant differences under four PBZ treatments (ANOVA, *p* ≤ 0.05).

**Figure 2 ijms-24-03753-f002:**
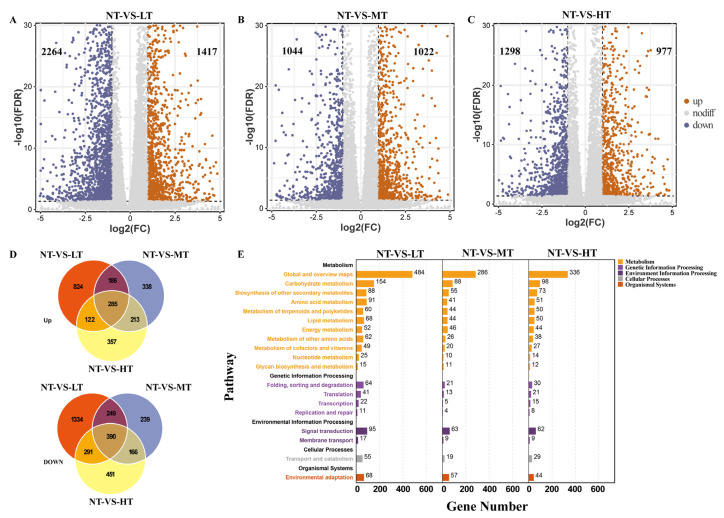
Transcriptomic responses in the roots of *P. bournei* with the different paclobutrazol treatments. (**A**–**C**) Summary of the numbers of up- and down-regulated differentially expressed genes (DEGs) in roots. (**D**) Venn diagrams showing the numbers of common and specific DEGs across four groups, including up-regulated DEGs and down-regulated DEGs in roots of *P. bournei*. (**E**) DEGs are significantly enriched in KEGG pathways of five biological functional processes. Signal transduction enriched the most DEGs in the environment information processing.

**Figure 3 ijms-24-03753-f003:**
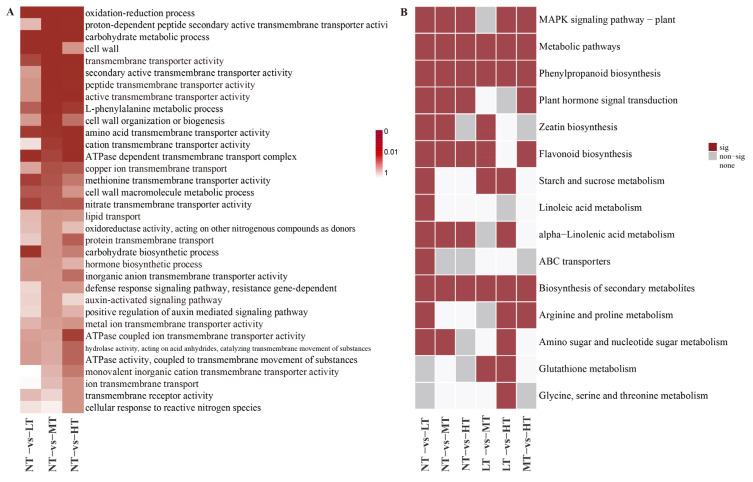
(**A**) Gene Ontology (GO) terms for the primary biological processes of the paclobutrazol responsive transcripts in the roots under different PBZ concentrations (false discovery rate (FDR) < 0.05). The plant hormones processes highlighted by blue words are those with higher enrichment under MT conditions than under other conditions. (**B**) Summary of the key KEGG pathways that are significantly enriched by various differentially expressed gene sets (false discovery rate (FDR) < 0.05). NT, LT, MT, and HT represent the control and three treatment groups, respectively.

**Figure 4 ijms-24-03753-f004:**
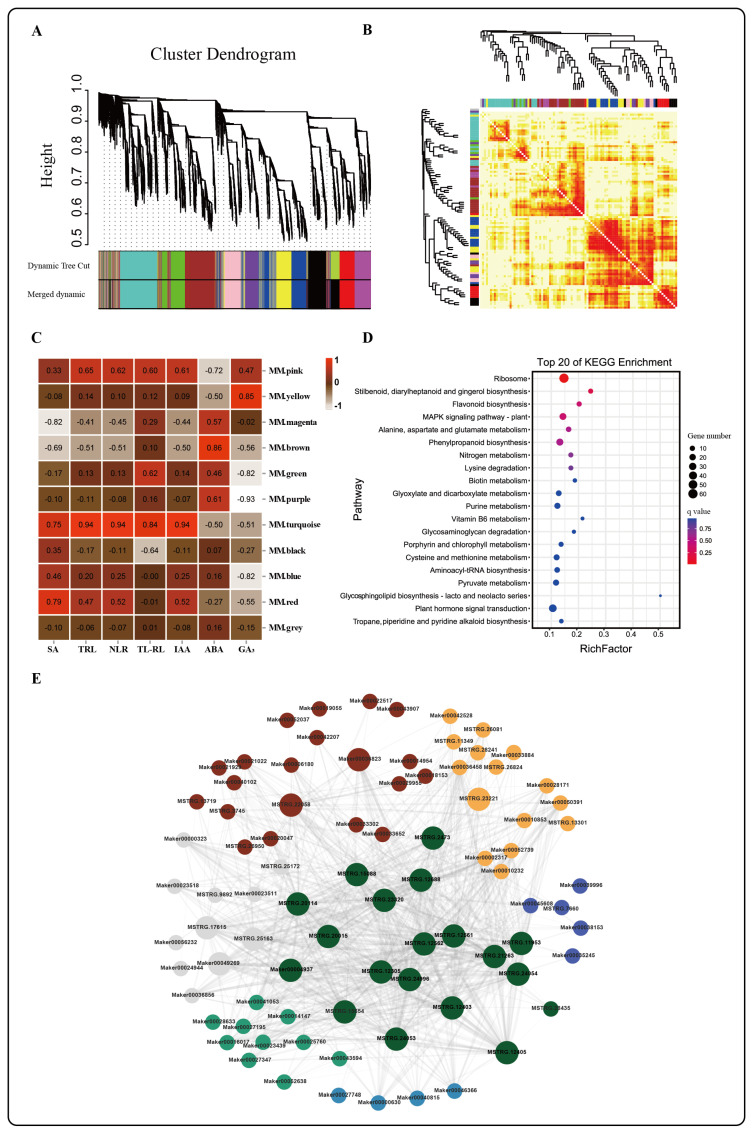
(**A**) Topological overlap matrix (TOM) heatmap showing the correlation among gene module memberships in the root of *P. bournei*. (**B**) Average linkage hierarchical clustering dendrograms of the genes. Modules, designated by color code, are the branches of the clustering tree. (**C**) Correlation analysis between a module and *P. bournei* treated by different concentrations of PBZ. The number of each cell is the correlation coefficient between each module gene and treated sample, and the colors of squares from red to white indicate the corresponding *p*-value (*p* < 0.05). (**D**) Significantly enriched KEGG pathways in the turquoise module. (**E**) Network regulation of the genes of interest. The hub genes are highlighted by the larger circle; the genes about root growth are gray; auxin, abscisic acid, and gibberellin-related genes are marked in red, turquoise, and cyan, respectively; transcriptions associated with the root are yellow and slate blue. SA, Total surface area, TRL, Total root length, NLR, number of lateral roots, TL-RL, Total length of the lateral root.

**Figure 5 ijms-24-03753-f005:**
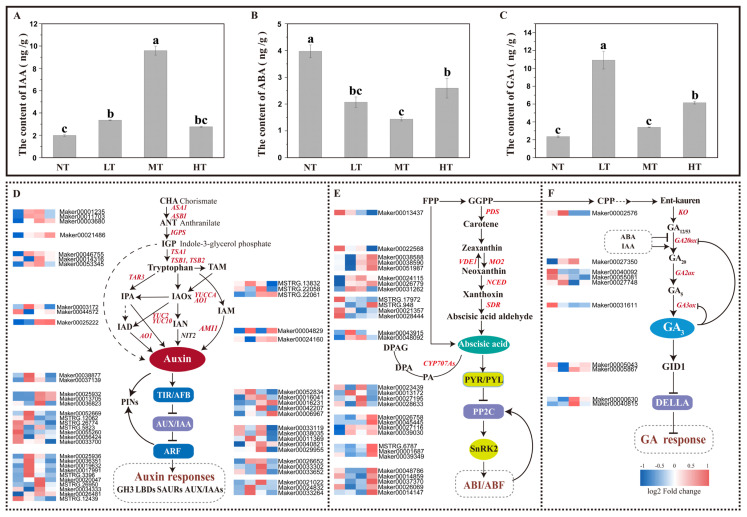
The concentration of (**A**) IAA, (**B**) ABA, and (**C**) GA_3_ in the roots of the indicated groups. Data are mean ± SD of three biological replicates. The letters indicate significant differences (ANOVA, *p* ≤ 0.05). (**D**–**F**) show the gene expression pattern of IAA, ABA, and GA_3_ biosynthesis and signaling in *P. bournei* roots from left to right.

**Figure 6 ijms-24-03753-f006:**
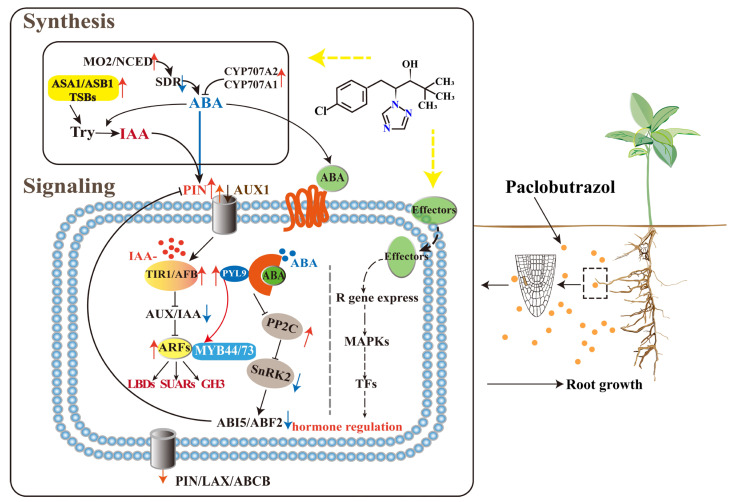
Possible molecular mechanisms of *P. bournei* roots in response to paclobutrazol based on hormonal regulation. Red and blue indicate up- and down-regulated phytohormones, respectively.

## Data Availability

The original sequence data have been deposited in the NCBI Sequence Read Archive (SRA) database under BioProject ID: PRJNA898267.

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
