# Peer review of "Paclobutrazol Promotes Root Development of Difficult-to-Root Plants by Coordinating Auxin and Abscisic Acid Signaling Pathways in Phoebe bournei"

_ijms, 2023, doi:10.3390/ijms24043753_

Round 1
Reviewer 1 Report
The manuscript is scientifically written in a detailed manner and the research subject is attractive. However, before acceptance, the manuscript should be checked for technical errors, grammatical, and other typographical errors. The references must be rechecked.
1- Rewrite the abstract, instead of a lengthy introduction, authors are requested to focus on results.
2- Authors performed qRT-PCR to validate some of the targets from the sequencing data. It is unclear from the method, whether identical or different biological RNA samples (number of independent biological samples and replicas) were used for sequencing and qRT-PCR experiment. Please, clarify that part in the method section.
3- Authors should prepare qRT-PCR graphs to accompany correlation graphs.
4- Although the current study is fascinating to the scientific community and the authors have collected a unique dataset using cutting-edge methodology, the authors did not write the articles concisely, as well the presentation of data is poor quality. For example, in the abstract, they did not write anything about the physiological response that they mentioned in the results, and there is no direct link to that physiological analysis. The manuscript requires a minor revision. If the author applied the mentioned comments, the manuscript would be accepted.
Author Response
Dear Reviewer
Thank you for your comments concerning our manuscript entitled “Paclobutrazol promotes root development of difficult-to-root plants by coordinating auxin and abscisic acid signaling pathways in Phoebe bournei” (Manuscript ID: ijms-2138678). Those comments are valuable and very helpful.
We have read those comments carefully and extensively modified the original manuscript, including the abstract, qRT-PCR graphs, and text formatting. We also briefly clarified the results, such as physiological data. The responses to the reviewer's comments are marked in red and presented following. We hope you will be satisfied with our answers and the new revision we provided.
We would love to thank you for allowing us to resubmit a revised copy of the manuscript and we highly appreciate your time and consideration.
With best regards

Reviewer 2 Report
Paclobutrazol promotes root development of difficult-to-root plants by coordinating auxin and abscisic acid signaling pathways in Phoebe bournei
Jing Li, Peiyue Xu, Yanyan Song, Shizhi Wen, Yujie Bai, Li Ji, Yong Lai, Gongxiu He, Dangquan Zhang
This is an interesting and novel study with important applications for rare and endangered endemic plant, in this case Phoebe bournei.
The manuscript provides new vegetative and molecular insights for solving rare plants' root growth problems through the use of Paclobutrazol, a plant growth regulator used in agriculture. In particular, the manuscript gives an interesting overview from a molecular point of view on the effects of Paclobutrazol on plants hormones signaling pathways.
The manuscript is well written, linear with the explanation of the analysis performed, and strictly focused on the role of hormones in plants. The figures are well drawn; statistics are applied correctly. The introduction and the discussion mention several original works on the topic, citing the latest findings. I just recommend standardizing the font size of several plants’ species names and genes abbreviations throughout the ms, as well as inserting spaces where they are missing especially in the introduction before the references numbers.
Author Response
Dear Reviewer
Thank you for your comments concerning our manuscript entitled “Paclobutrazol promotes root development of difficult-to-root plants by coordinating auxin and abscisic acid signaling pathways in Phoebe bournei” (Manuscript ID: ijms-2138678). Those comments are valuable and very helpful.
We have read those comments carefully and extensively modified the original manuscript. The responses to the reviewer's comments are marked in red and presented following. We hope you will be satisfied with our answers and the new revision we provided.
We would love to thank you for allowing us to resubmit a revised copy of the manuscript and we highly appreciate your time and consideration.
With best regards

Reviewer 3 Report
The manuscript “Paclobutrazol promotes root development of difficult-to-root plants by coordinating auxin and abscisic acid signaling pathways in Phoebe bournei” investigates the molecular mechanisms of different concentrations of PBZ on root growth of P. bournei. The results showed that PBZ treatments mediated the antagonist interaction of IAA and ABA to regulate the root growth in P. bournei. The research is innovative and has sufficient data support. However, Figures 2B and 2C are labeled the same (NT-VS-HT), which may be a mistake.
I believe that the publication of this manuscript will arouse the interest of researchers and have positive significance for the cultivation and protection of rare plants.
Author Response
Dear Reviewer,
Thank you for your comments concerning our manuscript entitled “Paclobutrazol promotes root development of difficult-to-root plants by coordinating auxin and abscisic acid signaling pathways in Phoebe bournei” (Manuscript ID: ijms-2138678). Those comments are valuable and very helpful.
We have read those comments carefully and extensively modified the original manuscript. The responses to the reviewer's comments are marked in red and presented following. We hope you will be satisfied with our answers and the new revision we provided.
We would love to thank you for allowing us to resubmit a revised copy of the manuscript and we highly appreciate your time and consideration.
With best regards
